# Type 1 Nuclear Receptor Activity in Breast Cancer: Translating Preclinical Insights to the Clinic

**DOI:** 10.3390/cancers13194972

**Published:** 2021-10-03

**Authors:** Sanjeev Kumar, Allegra Freelander, Elgene Lim

**Affiliations:** 1Faculty of Medicine, St Vincent’s Clinical School, University of New South Wales, Darlinghurst 2010, Australia; a.freelander@garvan.org.au (A.F.); e.lim@garvan.org.au (E.L.); 2Garvan Institute of Medical Research, University of New South Wales, Darlinghurst 2010, Australia

**Keywords:** nuclear receptor, steroid hormone, oestrogen, progesterone, androgen, glucocorticoid, breast cancer

## Abstract

**Simple Summary:**

Breast cancer is the most common cancer affecting women, and the importance of NR function in breast cancer biology has been recognized since the turn of the 20^th^ century. The nuclear receptor family of transcription factors is associated with cancer development and progression, informs diagnostic and prognostic outcomes, and is an established therapeutic target. Across all subtypes of breast cancer, crosstalk between NR pathways and other signalling pathways has also been demonstrated. Recent critical findings into modulating these NRs, particularly Type 1 NRs, have led to clinical trials and a greater understanding of their mechanism of action. Here, we reviewed the current preclinical insights into the role of Type 1 NRs in breast cancer that have served as a catalyst for clinical translation.

**Abstract:**

The nuclear receptor (NR) family of transcription factors is intimately associated with the development, progression and treatment of breast cancer. They are used diagnostically and prognostically, and crosstalk between nuclear receptor pathways and growth factor signalling has been demonstrated in all major subtypes of breast cancer. The majority of breast cancers are driven by estrogen receptor α (ER), and anti-estrogenic therapies remain the backbone of treatment, leading to clinically impactful improvements in patient outcomes. This serves as a blueprint for the development of therapies targeting other nuclear receptors. More recently, pivotal findings into modulating the progesterone (PR) and androgen receptors (AR), with accompanying mechanistic insights into NR crosstalk and interactions with other proliferative pathways, have led to clinical trials in all of the major breast cancer subtypes. A growing body of evidence now supports targeting other Type 1 nuclear receptors such as the glucocorticoid receptor (GR), as well as Type 2 NRs such as the vitamin D receptor (VDR). Here, we reviewed the existing preclinical insights into nuclear receptor activity in breast cancer, with a focus on Type 1 NRs. We also discussed the potential to translate these findings into improving patient outcomes.

## 1. Nuclear Receptors in Breast Cancer

An appreciation of the fundamental importance of nuclear receptor (NR) function in breast cancer biology can be initially traced back to Beatson’s pivotal work in 1896 demonstrating that oophorectomy in young women with unresectable breast cancer could cause tumour regression [1]. Fast-forward to the 1960s, Jensen followed by Toft and Gorski used murine models to confirm the existence of an intracellular hormone-binding entity that we now recognise as a subset of NR called the steroid hormone receptor [2]. Historically, the study of NR function in breast cancer has largely focused on characterising and therapeutically targeting the oestrogen (ER) and, to a lesser extent, the progesterone (PR) and androgen receptors (AR). However, recognising that ER/PR-negative breast cancers, and indeed a subset of ER/PR-positive breast cancers, do not benefit from ER-targeted therapies has provided a clinical imperative to evaluate the therapeutic impact of modulating non-oestrogen NR-signalling pathways as well. A list of the main nuclear receptors in breast cancer can be found in Table 1.

NRs are ligand-activated intracellular proteins or DNA-binding transcription factors, [3,4,5] that play a crucial role in key aspects of human physiology and pathophysiology, such as breast cancer. The NR family of transcription factors consists of 48 different genes, including the closely related steroid hormone receptors. Typically, nuclear hormone receptors consist of several domains, which share a similar structure and are differentially conserved between each receptor (Figure 1a). Typically, ligands (such as hormones, vitamins, or nutrients [6]) bind to their cognate NRs, which in turn regulate the transcription of a subset of genes expressed by a cell, resulting in what is typically considered ‘classical’ hormone signalling. Some NRs, however, can remain constitutively active. To enable these pathways and provide an added layer of regulation, NRs recruit protein coregulators into transcriptional complexes that bind to the DNA. These are either coactivators, which enhance transcription, or corepressors, which repress transcription [4]. These coregulators are essential to the underlying mechanism of NR action by mediating their transcriptional potency [7] and therefore modulating gene expression [8]. 

NRs are classically divided into two subsets based on their subcellular location and function [3]. Type 1 nuclear receptors include the sex steroid hormone receptors AR, ER and PR, as well as the corticosteroids GR and the mineralocorticoid receptor (MR). Broadly, sex steroids control the development of the urogenital tracts, secondary sexual characteristics and behaviour, as well as gametogenesis in both males and females. Corticosteroids control our physiological, developmental and behavioural responses to stress, as well as play a critical role in regulating salt and water balance. Unliganded, Type 1 NRs share a similar structure [6,9]. With classical direct genomic signalling, Type 1 NRs are located in the cytoplasm, but translocate to the nucleus upon hormone binding as homodimers, where they associate with chromatin and regulate gene transcription (Figure 1b). However, particularly with ER signalling, we now recognize that indirect regulation of gene expression can also occur via genomic, nongenomic and ligand-independent pathways, comprehensively reviewed by Fuentes et al. [10] and Siersbaek et al. [11]. These insights into the interplay between intracellular kinases, transcription and growth factors, membrane receptors, coregulators and both natural and synthetic ligands with NR signalling have helped in the design of practice-changing therapeutic strategies. In contrast, Type 2 NRs, which include the thyroid hormone, vitamin D and retinoic acid receptors, reside in the nucleus constitutively bound to DNA as RXR heterodimers, and typically act to repress transcription in the absence of ligands [3,12].

While this review focuses predominantly on recent preclinical insights into Type 1 NRs with translational potential in ER+ breast cancer, this is a rapidly evolving space in molecular biology and the clinic that is becoming increasingly relevant across all breast cancer subtypes.

## 2. Nuclear Receptor Autoregulation and Crosstalk

NR autoregulation describes classical single nuclear receptor activity, or the standard regulation of expression of a NR gene by its hormone-bound protein product [3]. The autoregulation of NR genes can lead to induction (upregulation) or repression (downregulation). Autoinduction leads to the cellular biosynthesis of more NRs and enhanced hormone responsiveness, while autorepression is a homeostatic mechanism that modulates hormonal signalling by downregulating the hormone receptor. Mechanisms for NR autoregulation can be broadly transcriptional or posttranscriptional.

In addition to ligands autoregulating the expression of their own receptor, another important means for modulating cellular responsiveness to different hormonal signals is through the cross-regulation of expression of other NRs. This ‘crosstalk’ between nuclear receptors can be defined as the interplay between different nuclear receptors or even between their overlapping signalling pathways [6]. DNA crosstalk mechanisms can be classified as either indirect (where NRs do not physically interact) or direct (where NRs physically interact to jointly regulate specific target gene subsets). Broadly, when these NR-induced signalling pathways either indirectly or directly interplay, this creates unique signalling and gene expression profiles. An understanding of the different NR crosstalk mechanisms provides us with novel opportunities to therapeutically target oncogenic pathways. For deeper mechanistic insights into co-regulators, nuclear receptor signalling and crosstalk, we direct the reader to other comprehensive reviews from Bagamasbad et al. [3], Conzen et al. [5], De Bosscher et al. [6], O’Malley et al. [7], Sikora et al. [8], and Doan et al. [13].

Biological insights into the underlying mechanisms of this complex NR interplay, both at the level of the cistrome and the interactome, have been gained from techniques such as Chromatin Immunoprecipitation coupled with high-throughput Sequencing (ChIP-seq) and Rapid Immunoprecipitation Mass spectrometry of Endogenous protein (RIME) [14,15,16,17,18]. How these insights impact the targeting of NRs as a therapeutic pathway in breast cancer, and highlight potential combinatorial strategies, are addressed in this review.

## 3. Oestrogen Receptor Signalling

The action of oestrogen is mediated by two ERs encoded on different chromosomes but sharing sequence homology, the Estrogen Receptor alpha (ERα, NR3A1) and Estrogen Receptor beta (ERβ, NR3A2) nuclear hormone receptors [19]. Meanwhile, the role of ERβ in breast cancer remains unclear and is not specifically assessed by immunohistochemistry in the clinical setting [20]. ERα has been intensely studied due to its integral role in breast tumorigenesis, where it can initiate gene expression changes that promote cell cycle progression [21]. In modern-day epidemiological studies, exposure to oestrogen (and progestogens) has been consistently linked to an increase in breast cancer risk [22], particularly ER+ breast cancer [23]. Around 75% of breast cancers are defined and driven by ERα transcriptional activity and for decades, anti-oestrogen therapies targeting ER have formed the cornerstone of therapy for the management of ER+ breast cancer. ERα inhibition is achieved either by the direct blockade of ERα activation through competitive inhibition of oestradiol (selective ER modulators, SERMS, i.e. tamoxifen), degrading ER (selective ER degraders, SERDs, i.e. fulvestrant) or by preventing peripheral oestrogen synthesis using aromatase inhibitors (AIs). However, clinical outcomes vary considerably, and a proportion of women with early breast cancer driven by ERα transcriptional activity develop drug resistance and relapse with incurable, metastatic disease. 

More recently, the targeting of cell cycle progression with cyclin-dependant kinase 4/6 (CDK4/6) inhibitors in combination with anti-oestrogen therapy has become the standard first-line therapy in de novo or recurrent breast cancer [24,25]. While this approach has prolonged the progression-free and overall survival of patients with metastatic disease [26,27,28,29], when patients develop resistance to this treatment combination, there are currently no equally robust therapeutic strategies in the second line [30]. Second-line treatment options include fulvestrant, cytotoxic chemotherapy, or anti-oestrogen therapy combined with inhibitors of the mammalian target of rapamycin (mTORi). Additionally, growing evidence supports the use of next-generation oral SERDs (Table 2) and combinatorial strategies with alpha-selective phosphoinositide 3-kinase inhibitors (PI3Ki) in patients harbouring somatic, activating PIK3CA mutations [31,32]. The optimal sequencing of this repertoire of therapeutic strategies, however, remains the subject of ongoing clinical trials such as the SONIA study (ClinicalTrials.gov identifier NCT03425838), and the growing use of commercially available tissue and liquid biopsy-based companion diagnostic panels is leading to a surge in biomarker-driven treatment strategies post-CDK4/6-inhibitor therapy in ER+ metastatic breast cancer.

As a member of the nuclear receptor superfamily of transcription factors, ERα is composed of functional domains and structural regions in common with other nuclear receptors (Figure 1a). The A/B region represents the N-terminal domain, involved in gene transcription transactivation, and containing a zinc finger that mediates binding to target sequences [10]. Significant progress has been made in understanding how wild-type ERα interacts with DNA. Using genomic technologies in breast cancer cell lines, it has been shown that ERα is able to bind to specific DNA sequences known as estrogen response elements (EREs, with a consensus motif GGTCAnnnTGACC) within the chromatin [33,34]. The C region of ERα is the DNA binding domain which contributes to ER dimerization and binds to these canonical EREs. The D region is the hinge region that binds chaperone proteins and allows for receptor–ligand complexes to translocate to the nucleus, and the E/F region is the ligand binding domain, which binds oestrogen, coactivators and corepressors (Figure 1a). Additional regulators of ER transcriptional activity known as activation function domains AF1 and AF2 are located in the N-terminal and DNA binding domains, respectively [35]. 

Genomic analyses have shown that ERα rarely acts through associations with promoter regions of target genes. Instead, genome-wide maps of ERα binding in breast cancer confirm that ERα binding events mostly occur at distal cis-regulatory enhancer elements at significant distances from the transcription start sites [33,36]. DNA-looping occurs, bringing enhancers in spatial proximity to promoter regions of target genes, and transcription is initiated [37]. It is also now clear that, in addition to oestrogens, ER function is modulated by other steroid receptors, with the interaction between ER and both PR and AR being the best-characterized models of nuclear receptor crosstalk in breast cancer. In addition, multiple signalling pathways (e.g., growth factor and cytokine signalling pathways) may also have a substantial impact on the efficacy of anti-oestrogen therapies [8,11].

It has been proposed that resistance to endocrine therapies may be the result of both genetic and epigenetic factors [38,39,40,41]. While gain-of-function mutations in *ESR1*, the gene encoding ERα, are relatively rare in primary breast cancer [42], 11–55% of metastatic cancers have point mutations in the ligand-binding domain of ER, especially in amino acids Y537 and D538 [42]. These mutations generate a constitutively active ER that is less dependent on oestrogen for activity [43,44,45]. The highest prevalence of *ESR1* mutations has been reported in the cell-free DNA (cfDNA) of AI-resistant metastatic ER+ breast cancer patients using droplet digital PCR [46,47,48]. At a cistromic level, not only are *ESR1* mutants distinct from oestrogen-stimulated wild-type ER, *Y537S* and *D538G ESR* mutants also have distinct cistromes and transcriptomes [49]. These mutations cluster in the ligand-binding domain of ER and lead to ligand-independent ER activity that promotes tumour growth, partial resistance to endocrine therapy and enhanced metastatic capacity. However, tumours bearing *ESR1* mutations can retain relative sensitivity to SERDs. A retrospective analysis of plasma samples from the phase III SoFEA study in patients resistant to nonsteroidal aromatase inhibitors confirmed significantly improved outcomes with fulvestrant-containing regimens compared with exemestane in the 39.1% of patients found to be harbouring an *ESR1* mutation [47]. This highlights the imperative to accelerate the development and rollout of more effective, potent and orally bioavailable next-generation SERDs, which have been shown to act through slowing intra-nuclear ER mobility, resulting in the limitation of both chromatin accessibility and downstream proliferative activity [50]. A number of these next-generation SERDs are currently being investigated in phase I–III clinical trials in early- and advanced-stage breast cancer, both alone and in combination with CDK4/6 inhibitors (ClinicalTrials.gov identifiers NCT04647487, NCT03455270, NCT04669587 and NCT04711252).

The novel selective estrogen receptor covalent antagonists (SERCAs) are another therapeutic alternative in overcoming endocrine resistance. This class of drugs targets the cysteine residue at amino acid 530 (C530) that exists only in ER, to promote a unique antagonist conformation. Specifically, the SERCAs H3B-5942 and H3B-6545 have been demonstrated to covalently bind to C530 of both wild-type and mutant ERα proteins, and have been shown to be superior to standard-of-care therapies in in vitro and in vivo models of endocrine resistance [51,52]. Currently, H3B-6545 is being investigated in phase I–II clinical trials in advanced, metastatic breast cancer either alone or in combination with CDK4/6 inhibitors (ClinicalTrials.gov identifiers NCT04568902, NCT04288089 and NCT03250676).

Emerging data have revealed that alterations in the epigenome can result in ER-directed therapy resistance. Aromatase inhibitor-induced DNA hypermethylation at estrogen-responsive elements (EREs) is associated with a reduction in ER binding and activity [41,53]. As a result, decreased gene expression of key ER regulators and reduced ER binding cause the cell to become less dependent on oestrogen for survival, and therefore less sensitive to ER-directed therapies [41]. Moreover, ER+ breast cancers that have relapsed following endocrine therapy exhibit higher DNA methylation at enhancer loci [41]. ER-directed therapy resistance can also emerge from altered methylation patterns at promoter regions of genes [54,55]. For example, hypermethylation at the *PTEN, PIXT2* and *HOXC10* promoters have been found to be predictive biomarkers of resistance in both cell lines and tumour tissues [55,56,57]. While demethylation therapies such as decitabine have been demonstrated to be efficacious in reversing hypermethylation in preclinical models, they have not yet been translated into clinical use [58]. 

Post-translational histone modifications have also been shown to induce chromatin remodelling that favour the repression of ER and promotion of signalling pathways associated with endocrine resistance. Histone variants have been linked to oestrogen signalling and endocrine resistance. For example, overexpression of the H2A variant H2A.Z has been linked to oestrogen-independent proliferation, and thus ER-targeted therapy resistance [59]. Another study demonstrated that the H2B variant HIST1H2BE is overexpressed in both resistant cell lines and tumours treated with aromatase inhibitors derived from patients [60]. Finally, variations in histones by histone deacetylases (HDACs) have also been associated with the loss of ER expression, also conferring endocrine resistance. In early-phase clinical trials, combinations of an aromatase inhibitor or tamoxifen with HDAC inhibitors such as vorinostat or entinostat demonstrated improvements in overall survival, with the potential to re-sensitize tumours to endocrine therapy in women with resistant disease [61,62]. Disappointingly, results from the positive phase II ENCORE301 study were not replicated in the phase III E2112 study of entinostat plus exemestane, which failed to demonstrate improved survival compared to exemestane alone in aromatase inhibitor-resistant patients with advanced disease [63]. Unlike ENCORE301, many patients in the E2112 study had received prior fulvestrant and/or CDK4/6-inhibitors, likely impacting the final outcome, while simultaneously strengthening the trial’s relevance in a more ‘current’ therapeutic context where CDK4/6-inhibitors are now the standard first-line treatment. We are awaiting results from exploratory analyses to identify a predictive biomarker of response to this class of therapy.

## 4. Progesterone Receptor Signalling

In combination with oestrogen, progesterone plays a major role in normal breast development, as well as changes in the mammary gland during the menstrual cycle, pregnancy and lactation [64]. Mouse mammary studies have revealed that PR (NR3C3) is essential in the mammary epithelium for ductal side-branching and alveologenesis [65]. In the adult mouse, 17β-oestradiol induces the expression of PR, and stimulation with progesterone (e.g., during the luteal phase of the menstrual cycle) triggers cell proliferation. Progestogens, which are any natural or artificial substance that exerts progesterone-like activity via activation of PR [66], have long been employed as contraceptive agents, or components of Menopausal Hormonal Therapy (MHT) in combination with oestrogen as they prevent hyperplastic or malignant consequences of chronic, unopposed oestrogen exposure on the endometrium [67]. 

PR is both a member of the NR family and an ERα target gene, co-expressed in over two-thirds of ER+ breast cancers [68]. *PR* exists in two isoforms, PR-A and PR-B [69]. Historically, the accepted explanation for PR activity in ER+ breast cancer cells was that PR expression was a passive consequence of a functional oestrogen receptor. PR was therefore established as a biomarker of ERα functionality in breast cancer and a predictive marker of response to ERα-directed agents [70]. Therefore, functional studies into the role of progesterone and its receptor have lagged significantly behind those of ERα and have in fact been the subject of heated debate. Epidemiological studies and clinical MHT trials have implicated that synthetic progestins were associated with an increased risk of breast cancer [22,23,71,72]. Additionally, progesterone-initiated PR signalling has been shown to contribute to mammary tumourigenesis in murine models [73]. On the other hand, multiple studies have demonstrated the improved prognosis of PR+ breast cancers [74,75,76,77,78,79], and a Combined Endocrine Receptor (CER) score averaging the Allred score of both ER and PR has been demonstrated to be a more powerful discriminator of patient outcome than either ER or PR alone [80]. In support of this, the heterozygous or homozygous deletion of PR occurs more often in the luminal B breast cancer subtype that is associated with a higher proliferation rate and poorer prognosis compared with luminal A breast cancers [81].

Importantly, the role of PR in breast cancer is context-dependent and highlights the importance of the hormonal milieu. Pivotal studies have emphasised that the function of PR in breast cancer has to be considered in the context of the presence of oestrogen and ERα signalling [15,82]. In the absence of a functional oestrogen-activated ER complex, PR activation might have modest pro-proliferative effects, and differing effects in malignant and normal breast tissue contingent on oestrogenic status. This emphasizes the importance of further delineating the precise mechanisms through which PR regulates tumours compared with mammary gland proliferation.

In breast cancer cell lines, the administration of progestogens has been shown to inhibit ERα transcriptional activity and oestrogen-induced cell proliferation [83,84,85]. Additionally, progestogens alone oppose the oestrogen-induced proliferation of MCF7 and T47D cell line and ER+/PR+ patient-derived xenografts [15,86]. Interestingly, the combination of tamoxifen and progesterone had an even greater suppressive effect on tumour growth. Clinically, benefits have been demonstrated from a single injection of progesterone administered before surgery [87], and the use of a single agent progestogen has consistently shown to be clinically beneficial either as a first-line therapy in de novo metastatic ER+ breast cancer, or in advanced disease when ER-directed endocrine agents have failed [88,89,90,91,92,93,94,95,96,97,98,99,100]. 

Taken together, existing data imply that PR can play an anti-proliferative role in ER+ breast cancer. Mechanistically, the stimulation of PR by progestins regulates distinct cistromes and transcriptomes in breast cancer cells compared to normal breast cells [101,102], and can function as a molecular rheostat to control ERα binding and transcriptional activity. In the presence of agonist ligands, insights gained from ChIP-seq experiments have confirmed that PR causes the rapid redistribution or sequestration of ERα away from its pro-proliferative gene targets in breast cancer cells, resulting in a unique gene expression program that is associated with a good clinical outcome and culminating in cell cycle arrest [15,103]. The potential clinical significance of exploiting this interaction between ERα and PR signalling in breast cancer affords the possibility that the addition of a progesterone agonist might enhance the anti-proliferative effect of anti-oestrogen therapies and therefore provide a more effective combination therapy. Indeed, two short-term, pre-operative window-of-opportunity studies, PIONEER and WinPRO (ClinicalTrials.gov identifiers NCT03306472 and NCT03906669), are currently testing this hypothesis clinically in early-stage breast cancer. 

Paradoxically, PR antagonists have also proven to be antiproliferative in cell line and murine models of ER+ breast cancer [104,105,106,107,108,109,110,111]. It has indeed been hypothesized that PR antagonists may also interfere with ER transcriptional activity, similarly to agonists [66]. However, clinical trials of agents such as mifepristone and onapristone have recruited poorly, have shown either a lack of reproducible efficacy or unacceptable hepatotoxicity [112,113,114], and have therefore not progressed to routine clinical use. 

## 5. Androgen Receptor Signalling

AR (NR3C4) is essential for the development of male reproductive organs, and it is also expressed in the majority of breast cancers. Two isoforms and several alternative splicing variants, encoded by the same gene, have been described. While AR is currently not routinely assessed immunohistochemically in biopsies and surgical excision specimens from our breast cancer patients, it is in fact the most widely expressed hormone receptor in all stages of breast cancer [115,116]. AR expression varies between breast cancer subtypes, and the function of AR in breast cancer is highly context-dependant, contingent on the co-expression of ER, the AR:ER protein ratio, the menopausal status of the patient and the hormonal milieu [8]. 

AR is expressed in up to 85% of ER+ breast cancer and is an independent clinico-pathological prognostic factor associated with favourable outcomes in this setting [117,118]. AR and ER co-localize at select genomic loci within the nuclei of breast cancer cells [119], and functional crosstalk between the hormone receptors has been well-described [120]. Controversially, both AR agonists and antagonists have been shown to inhibit growth in ER+ preclinical models, by inhibiting ER function at a genomic level [119,121,122,123,124]. However, the bulk of positive evidence supports an anti-proliferative role of androgens in ER+ breast cancers, with androgenic signalling via AR generally antagonistic of oestrogen activity [116,119,125]. Indeed, the historic use of androgens as a treatment in breast cancer clinically supports this [126,127]. Major reasons that have limited its clinical utility include virilising side effects, concerns regarding the aromatisation of androgens to oestrogen and the development of effective ER-directed approaches. 

More recently, there has been a resurgence of interest in the role of the AR signalling axis in ER+ breast cancer. The activation of AR by its natural ligand DHT in endocrine-sensitive ER+ cell lines in vitro inhibited proliferation and ER signalling [119,128], consistent with aforementioned clinical studies, supporting the premise that androgens inhibit proliferation and induce tumour regression. AR agonism has also been shown to retain its anti-ER signalling and growth inhibitory effects in endocrine therapy-naïve and -resistant in vivo patient-derived xenograft (PDX) models [129], including those harbouring genomic aberrations of *ESR1* and *CCND1* [16]. The anticancer effect was a result of an upregulation of AR target genes, including tumour suppressors, the reprogramming of the binding of ER and its co-activators on chromatin, and the redistribution of E2-stimulated p300 binding sites, resulting in inhibition of the expression of critical ER-regulated cell cycle and survival genes [16]. 

Therapeutically, the development of selective androgen receptor modulators (SARMs), such as enobosarm and RAD140, now offers a novel approach for targeting AR in breast cancer [16,130]. SARMs exhibit a high specificity of binding to AR and have the advantage that they dissociate the anabolic from androgenic effects of AR and therefore lack the virilising effects seen with the historical use of androgens [131]. A phase II study has demonstrated that enobosarm was well tolerated and conferred clinical benefit in heavily pre-treated patients with ER+/AR+ metastatic breast cancer [132]. Enobosarm in combination with CDK4/6 inhibitors has also been shown to be efficacious in the setting of endocrine or CDK4/6 inhibitor-resistant breast cancer models (which retain the expression of AR), suggesting that AR agonism may partially restore sensitivity to CDK4/6 inhibitors, thus positing that this combination may be an effective therapeutic strategy in the second-line treatment of ER+ metastatic breast cancer [16]. Together, these findings provided the motivation behind the phase III registration ARTEST study (ClinicalTrials.gov identifier NCT04869943) of enobosarm in endocrine therapy and CDK4/6 inhibitor-resistant ER+ metastatic breast cancer patients [133].

A transdermal preparation of 4-OH-testosterone, CR1447, has also demonstrated efficacy and a favourable toxicity profile in phase I/II studies [134]. Another alternative AR agonist includes oral testosterone undecanoate, which utilises a self-emulsifying drug delivery system to minimise pharmacokinetic variability experienced with other formulations [135], maximising free testosterone levels and AR interaction while potentially reducing the risk of androgenisation by more predictably maintaining testosterone levels in the eugonadal range [136]. Window-of-opportunity studies using this compound are also currently under development.

In contrast to ER+ breast cancer, high AR expression is associated with a poor prognosis in triple-negative breast cancers (TNBC) [137]. AR is expressed in 15–35% of TNBCs [138], primarily in the luminal androgen receptor (LAR) subtype. These tumours demonstrate a gene expression signature resembling that of endocrine-responsive tumours [139], are characterised by both AR expression and androgen-dependant growth [140], and are classically less responsive to conventional cytotoxic chemotherapy. AR-antagonism in this setting, with antiandrogens such as bicalutamide, have been investigated in clinical trials [141]. The second-generation AR antagonist enzalutamide has been shown to be well-tolerated in combination with exemestane, with clinical efficacy in this subset of TNBC that expresses AR [142], as well as in a cohort of ER+ patients with high levels of AR mRNA and low levels of ER mRNA on mRNA-sequencing [143]. This provides a rationale to explore combinatorial strategies of AR antagonists with agents such as CDK4/6i [144,145] and PI3K inhibitors [146]. Growing evidence also supports the utility of other next-generation AR-targeted agents such as abiraterone acetate with prednisone in AR+ TNBC, as well as the novel CYP17 lyase inhibitor and potent AR antagonist seviteronel (VT-464/INO-464), which is about to enter clinical trials in combination with chemotherapy in the phase I/II 4CAST study (ClinicalTrials.gov identifier NCT04947189). Furthermore, seviteronel has been shown to induce DNA damage following radiation in AR+ TNBC models, demonstrating a unique radiosensitizing effect [147]. 

## 6. Glucocorticoid Receptor Signalling

As with AR, GR (NR3C1) expression is not routinely evaluated in breast carcinomas. GR is expressed in ~70% of ER+ breast cancers and >60% of all breast cancers [148]. Clinically, synthetic glucocorticoids (GR agonists) such as dexamethasone are ubiquitously used in breast cancer in conjunction with chemotherapy to mitigate hypersensitivity reactions, and for their antiemetic, anti-inflammatory, as well as orexigenic properties. However, dexamethasone triggers different effects depending on the breast cancer subtype [149]. 

Similarly to the trend observed with AR expression, retrospective meta-analyses of patients with ER+ breast cancer have determined that high GR mRNA levels are associated with low tumour grade [148] and better prognosis [150,151] compared to low or negligible GR expression, independent of PR expression. Again, the opposite was observed in ER- breast cancer where high tumour GR expression was associated with a worse prognosis [150]. Therefore, mirroring AR, the tumour suppressor vs. oncogenic potential of GR is dependent upon ER expression. Mechanistically in ER+ disease, both ER and GR undergo crosstalk upon co-treatment with oestradiol and dexamethasone [151,152], with reciprocal reprogramming of both receptors occurring via the Hager lab’s well-described assisted loading mechanism [152,153]. This model involves chromatin remodelling by one receptor followed by recruitment of the other receptor. Upon co-stimulation of ER and GR in MCF7 cells with oestradiol and glucocorticoids, transcriptional changes in genes linked to cell proliferation and differentiation occur [151], inhibiting growth. Despite these preclinical findings, clinical studies in ER+ breast cancer patients have demonstrated varied effects of glucocorticoid use on breast cancer patient survival, with modest effects when used as a single agent and no additive effect in combination with other drugs, including anti-oestrogens [154,155]. However, given the widespread use of glucocorticoids in cancer, further exploration of the full therapeutic potential of activating GR in ER+ breast cancer warrants further investigation.

By contrast, GR activation with glucocorticoid has been shown to inhibit chemotherapy-induced cell apoptosis [156], and drives the expression of pro-tumorigenic genes in TNBC [157,158]. This raises the concern that the routine administration of high doses of synthetic glucocorticoids as a chemotherapy premedication has the potential to activate GR-mediated cell survival pathways and diminish the effectiveness of chemotherapy in this setting. More recently, an extensive study using both patient-derived and TNBC cell line-derived xenograft models also suggested that activation of GR with glucocorticoid treatment increases tumour heterogeneity and promotes breast cancer metastasis [159]. The suggested mechanism was the upregulation of the expression of a receptor tyrosine kinase-like orphan receptor-1 (ROR-1). With GR identified as a target in this setting, preclinical findings of the potentiated cytotoxic efficacy of paclitaxel chemotherapy with the addition of the GR antagonist Mifepristone [156] has now led to the translation of these findings to the clinical research setting. Consistent with these observations, a randomised phase I clinical trial combined mifepristone with nab-paclitaxel, confirming disease activity and manageable toxicity [160], and a randomised phase II trial is currently recruiting (NCT02788981).

## 7. Other Nuclear Receptors

The mineralocorticoid receptor (MR, NR3C2) is another receptor of interest, given that it is most likely expressed in the majority of breast cancers (up to 90%) [161]. Aldosterone, the primary ligand for MR, is typically used in the management of hypertension and cardiac failure. As with GR, MR can undergo crosstalk with PR to induce significant growth inhibition, and many glucocorticoids also bind to MR with high affinity [162]. An interaction between ER and MR has not been directly explored in breast cancer models; however, high MR and retinoic acid receptor (RAR) expression is associated with improved ER+ breast cancer-specific survival. A tumour-suppressive relationship between these nuclear receptors was illustrated by co-treatment with mineralocorticoids and retinoic acid receptor-stimulating retinoids [163], highlighting another potential therapeutic pathway worth characterising in the setting of breast cancer resistant to standard ER-targeted therapies. Independently, RAR expression can predict for resistance to tamoxifen [164]. While the results of combinatorial therapeutic strategies with retinoids in patients with breast cancer have been generally disappointing [165], retinoids have been shown to inhibit the expansion of chemoresistant cytokeratin 5-positive (CK5+) cells through RAR/PR crosstalk [166]. 

The relationship between Vitamin D and breast cancer remains controversial. A recent meta-analysis demonstrated that while there is no relationship between nuclear VDR (NR1I1) expression and overall survival in patients with breast cancer, high total nuclear and cytoplasmic VDR expression was associated with improved survival outcomes [167]. Both the VDR and vitamin D 1-hydroxylase, the enzyme that generates the active Vitamin D3 ligand 1,25-dihydroxycholecalciferol, are expressed in the human breast. Additionally, VDR expression correlates with the expression of ER. Interest has been provoked by growing evidence suggesting that adequate Vitamin D levels and intake inversely correlate with breast cancer risk [168,169,170], and that low levels of vitamin D are associated with an increased risk of recurrence or death in breast cancer patients [167,171]. This is arguably enough evidence to suggest that early breast cancer patients should at least supplement vitamin D if found to be deficient. 1,25-dihydroxycholecalciferol inhibits the proliferation of breast cancer cell lines and promotes differentiation and apoptosis in vitro [172,173,174], but clinical data regarding tumour responsiveness to vitamin D are limited and remain inconclusive [174,175]. The impact of vitamin D supplementation in the neoadjuvant setting on the rate of pathological complete response is being investigated in TNBC (ClinicalTrials.gov identifier NCT04677816).

## 8. Conclusions

Given that the prominent nuclear receptors in breast cancer all share similar consensus DNA binding motifs, as well as mechanisms of co-activation, it is not surprising that there is substantial functional crosstalk between these receptors. Additionally, given that most nuclear receptors are regulated by ligands and are often highly co-expressed, this renders them susceptible to external control over their gene regulatory activities [6], and therefore ideal druggable targets. 

In ER+ breast cancer, steroid hormone receptors are certainly not bystanders in ER signalling pathways. Modulation of their activity can alter or reprogram ER DNA binding to dramatically modify target gene expression. Such a modulation of hormone receptor function in combination with anti-oestrogen therapy can either modify the response or alter the trajectory of developing resistance to therapy, and certainly highlights the imperative to improve our understanding of how other steroid hormone receptors influence ER function in the context of standard anti-oestrogen therapy. However, the fact that AR and GR have opposite functions in the presence and absence of ER highlights the importance of characterising the functional interplay between different steroid hormone receptor signalling pathways in both luminal and nonluminal breast cancer subtypes, to fully exploit their therapeutic potential. Overwhelmingly, the modulation of nuclear receptor activity beyond targeting the ER provides us with novel approaches to manage patients with breast cancer.

Given the crosstalk of pathways triggered between different nuclear receptors, it may be possible that in the near future, breast cancer therapeutic decisions may require consideration of the expression of all four major steroid hormone receptors in breast cancer— ER, PR, AR and GR. This opens the prospect of the deeper characterisation of breast cancers for both prognostic purposes and as predictive biomarkers of response to a new array of endocrine therapies.

## Figures and Tables

**Figure 1 cancers-13-04972-f001:**
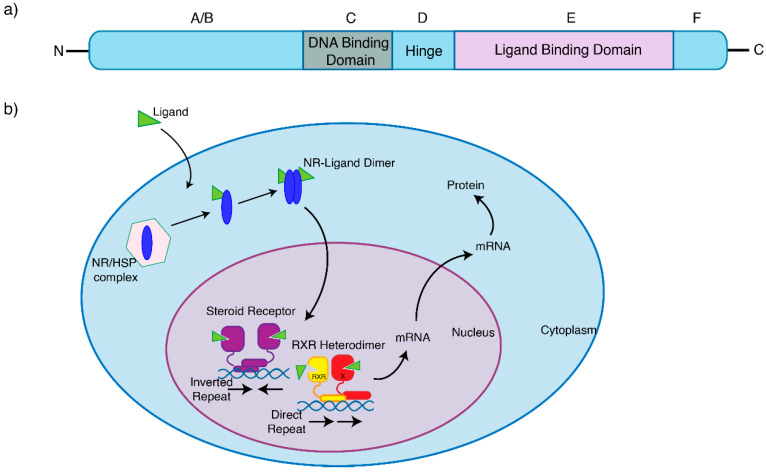
Nuclear receptors share the similar structural domains. (**a**) Most NRs contain an N-terminal region (A/B), a conserved DNA-binding domain (C), a variable hinge (D), a conserved ligand-binding domain (E), and a variable C-terminal region (F). (**b**) The NRs involved in breast cancer are bound to heat-shock proteins (HSP) in the cytoplasm, and are activated by binding of ligands, whereby canonical signalling is initiated. This NR–ligand complex will either form a heterodimer (steroid receptors), or form an RXR heterodimer, and translocate to the nucleus, binding to target genes and initiating transcription.

**Table 1 cancers-13-04972-t001:** Main nuclear receptors involved in breast cancer and their natural ligands.

Receptor	Receptor Type	Encoding Gene	Ligand
Oestrogen Receptor	Steroid Hormone	*ESR1*	Oestradiol, oestrogens
		*ESR2*	Oestradiol, 5α-androstane-3β,17β-diol
Progesterone Receptor	Steroid Hormone	*PGR*	Progesterone, progestogens
Androgen Receptor	Steroid Hormone	*AR*	Androgens, 5α-dihydrotestosterone (DHT)
Glucocorticoid Receptor	Steroid Hormone	*NR3C1*	Glucocorticoids, cortisols
Mineralocorticoid Receptor	Steroid Hormone	*NR3C2*	Aldosterone
Retinoic Acid Receptor	RXR Heterodimer	*RARA*	All-trans retinoic acid, 9-cis retinoic acid
		*RARB*	
		*RARG*	
Vitamin D Receptor	RXR Heterodimer	*VDR*	Calcitriol, 1,25-dihydroxy vitamin D_3_

**Table 2 cancers-13-04972-t002:** Clinical trials in progress that are investigating nuclear receptor-directed therapies in breast cancer.

NR Target	Treatment	Class	Combination Treatment	Phase	Stage	BC Subtype	ClinicalTrials.gov Identifier
**ER**	Giredestrant	Oral SERD	N/A	II	Neoadjuvant	ER+, HER2-	NCT04436744
			N/A	II	Advanced		NCT04576455
			Palbociclib	III			NCT04546009
	LY3484356		Abemaciclib, Trastuzumab, Alpelisib, Everolimus	I	Advanced	ER+, HER2-	NCT04188548
	Rintodestrant		Palbociclib	I	Advanced	ER+, HER2-	NCT03455270
	ZB716		Palbociclib	I, II	Advanced	ER+, HER2-	NCT04669587
	Camizestrant		Palbociclib	III	Advanced	ER+, HER2-	NCT04711252
	H3B-6545	SERCA	N/A	I	Advanced	ER+, HER2-	NCT04568902
			Palbociclib	I			NCT04288089
			N/A	I, II			NCT03250676
**PR**	Megestrol acetate	PR Agonist	Letrozole	II	Early, Window	ER+, HER2-	NCT03306472
	Prometrium		Letrozole	II	Early, Window	ER+, PR+, HER2-	NCT03906669
**AR**	Enobosarm	SARM	N/A	III	Advanced	ER+, AR+, HER2-	NCT04869943
	Severitonel-D	AR Antagonist	Docetaxel	I, II	Advanced	AR+, TNBC	NCT04947189
	Enzalutamide		N/A	II	Advanced	AR+, TNBC	NCT01889238
	Darolutamide		Capecitabine	II	Advanced	AR+, TNBC	NCT03383679
**GR**	Mifepristone	GR Antagonist	Nab-Paclitaxel	II	Advanced	GR+, TNBC	NCT02788981
			Pembrolizumab	II			NCT03225547
			N/A	II	Prevention	*BRCA1/2*^mut^ TNBC	NCT01898312
**VDR**	Vitamin D3	VDR Agonist	N/A	II	Advanced	TNBC	NCT04677816

NR: nuclear receptor; BC: breast cancer; ER: estrogen receptor; PR: progesterone receptor; AR: androgen receptor; GR: growth receptor; VDR: vitamin D receptor; SERD: selective estrogen receptor degrader; SERCA: selective estrogen receptor covalent antagonist; SARM: selective androgen receptor modulator; LHRH: luteinising hormone-releasing hormone; HER2: human epidermal receptor 2; BRCA1/2: breast cancer gene 1/2; TNBC: triple-negative breast cancer.

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
