# Peer review of "Type 1 Nuclear Receptor Activity in Breast Cancer: Translating Preclinical Insights to the Clinic"

_cancers, 2021, doi:10.3390/cancers13194972_

Round 1

Reviewer 1 Report

Thank you for theimprovements made.

Author Response

Many thanks for your input into our review article.

Reviewer 2 Report

The authors addressed effectively all comments. It is only necessary to remove the "(Ref?)" highlighted in yellow at page 6, line 182, and add the proper reference. 

Author Response

Many thanks for your input. This reference error has been duly amended. Thank you for highlighting this

This manuscript is a resubmission of an earlier submission. The following is a list of the peer review reports and author responses from that submission.

Round 1

Reviewer 1 Report

The topic of nuclear receptors and breast cancers is of great importance.

The authors decided to focus on the topic of translating preclinical insights into the clinic.

The reviewer has some experience in the field and understands how large this field is. The title of the manuscript is misleading as the review is focusing on a very small subset of Type 1 NR-s, and this needs to be specified from the beginning.  We understand that at point 7 other nuclear receptors are mentioned but that part is so short that we suggest to disregard completely and focus only on ER/PR/AR named also sex hormone receptors.

The presented insights are relevant and important, albeit the reviewer would like to emphasize some aspects that would need further elaboration or are not acceptable.

Page 2 regarding the location of Type 1 NR-s in the cytoplasm. 

While this was the general concept for decades there is a tremendous body of evidence at least for ERa that it is presently attached to the DNA on specific enhancers even before the cells are treated with estrogen.  Please review the evidence on this line and present how our understanding has changed.

Figure 1: quite schematic please show at least the binding sites as Inverted repeats (IR-s) and Direct Repeats (DR-s).

Huge topics about how these receptors work are missing therefore the interpretation of the clinical aspects are not explained in detail. Such topics are: (1) the consensus sequence of the binding sites and their spacing and orientation, (2) the structure of the Ligand-binding domain and the functional importance of different helixes e.g. in the binding of tamoxifen or different SERMS, (3)post-translation modifications of the NR-s especially of the estrogen receptors (not histones!) and their relevance in the light of different ligands used in clinical trials, (4) how do the ligands activate the receptors e.g how the binding of the various ligands can modify the recruitment of different coactivator proteins, (5) degradation of NR-s after ligand binding, (6) non-genomic effects of NR-s e.g. the large body of evidence about the non-genomic mechanisms of estrogen receptor alpha and the importance of these effects in potential clinical applications. 

A dramatic error is the title of point 6. named Growth Receptor Signalling. 

We would emphasize the importance of using the systematic names of the nuclear receptors through the whole manuscript together with their accepted names. These systematic names such as NR3A1 for ERapha would prevent the exchange of glucocorticoid receptors with growth hormone receptors that are peptide receptors.

The reviewer would recommend to disregard this review and write maybe a different one with a more focused topic e.g. only a subgroup of NR-s and the clinical trials of that narrow group. Another suggestion would be to involve some experts with a deeper understanding of the molecular details of how NR-s work.  A third suggestion would be to address the reader to some well-writen reviews about specific topics not described in this manuscript.

Good luck!

Reviewer 2 Report

In this review article “Nuclear Receptor Activity in Breast Cancer: Translating Pre-clinical Insights to the Clinic”, Kumar and colleagues summarized the current understanding of the nuclear receptor family of transcription factors, including ER, PR, AR, GR, MR and VDR, in the development, progression, and treatment of breast cancer.  The review did not focus on the detailed molecular mechanisms, but centered on how the interaction among ER and other nuclear receptors may be used to improve patient outcome.

Some suggestions are listed below:

  1. In Table 1, the authors should consider to add another column to describe whether the receptor is pro- or anti-tumorigenic/proliferation/metastasis/recurrence in breast cancer.
  2. Figure 1 can be improved. The authors should consider to include information of cellular localization of the NRs with or without ligands and how ligands affect steroid receptors and RXR heterodimer into the figure.
  3. The authors should consider to make a summary figure describing the interaction between these NRs and their crosstalk in breast cancer development and progression. This summary figure will help the readers to grasp the idea why this NR network is important for breast cancer therapies.

Minor points:

  1. The text font should be consistent. There are two different text fonts on page 3 and 4.
  2. On page 4, line 155-156: “increased ER turnover, resulting in limited chromatin accessibility and downstream anti-proliferative ” Should change to “, resulting in limited chromatin accessibility and downstream proliferative activity.”

Reviewer 3 Report

The authors provide a well-written, comprehensive review on the role of nuclear receptors in breast cancer, delivering a good and easy-to-read resumen of the state-of-the-art in this field, with educational merits worth acknowledging. However, I have some concerns that I would like the authors to address before recommending publication: 

Major

1) Page 4, lines 121-122, instead of citing 23,24, which are not appropriate for the concept expressed in the sentence, I would slightly modify the ending by hilighting the role for mTOR inhibitors and PI3K inhibitors in this subset, and would cite the following as references: PMID: 33794206, 33909934, and 33810205.

At the same time it should be stressed that sequencing trials are needed to define optimal therapeutic sequencing. Cite the ongoing SONIA trial NCT03425838.  Please comment also on the need for biomarkers guiding more personalised treatment choices after CDK4/6i inhibitors, a context where we still don’t know if CT, for example, might also play a role, compared to other ET options. 

2) At the end of section III Entinostat phase III trial's failure has to be mentioned: Connolly RM, Zhao F, Miller KD, et al. E2112: Randomized phase III trial of endocrine therapy plus entinostat or placebo in hormone receptor–positive advanced breast cancer. A trial of the ECOG-ACRIN cancer research group. J Clin Oncol. Published August 6, 2021. doi:10.1200/JCO.21.00944

3) In the final part of the section V, reference 138 (Krop Clin Can Res 2020) is not appropriate, since the population in that study was affected by HR+ BC. However, results from this study with respect to AR mRNA levels and ESR1 mRNA levels makes it worth mentioning. Please reshape the section differentiating between TNBC and HR+/HER2-neg MBC. 

Minor

1) Page 4, line 120, after “with metastatic disease” cite these two systematic reviews and meta-analyses that provide comprehensive assessments of PFS and OS for CDK4/6 combos: PMID: 31859246 and PMID: 32407488.

2) Page 9, line 372, the statement “has now led to translation of these findings to the clinic.” might be misinterpreted. I would suggest the authors to replace "to the clinic" with "to the clinical research stage".